# Epicardial electrical heterogeneity after amiodarone treatment increases vulnerability to ventricular arrhythmias under therapeutic hypothermia

Chin-Yu Lin[1,2], Ting-Yung Chang[1,2], Yu-Feng Hu[1,2,3]*, Yu-Cheng Hsieh[4], Yi-Jen Chen[5], Hung-I Yeh[6], Yenn-Jiang Lin[1,2], Shih-Lin Chang[1,2], Li-Wei Lo[1,2], Tze-Fan Chao[1,2], Fa-Po Chung[1,2], Jo-Nan Liao[1,2], Ta-Chuan Tuan[1,2], Shih-Ann Chen[1,2,4,7]*

1 Department of Medicine, National Yang Ming Chiao Tung University, Hsinchu, Taiwan, 2 Department of Heart Rhythm Center, Taipei Veterans General Hospital, Taipei, Taiwan, 3 Institute of Biomedical Sciences, Academia Sinica, Taipei, Taiwan, 4 Department of Cardiovascular Center, Taichung Veterans General Hospital, Taichung, Taiwan, 5 Division of Cardiovascular Medicine, Department of Internal Medicine, Wan Fang Hospital, Taipei Medical University, Taipei, Taiwan, 6 Department of Medicine, Mackay Medical College, Taipei, Taiwan, 7 Department of Medicine, National Chung Hsing University, Taichung, Taiwan

* huhuhu0609@gmail.com (YFH); epsachen@ms41.hinet.net (SAC)

**Data Availability Statement:** All relevant data are within the paper and its Supporting Information files.

## Abstract

### Background

Amiodarone is commonly used during therapeutic hypothermia (TH) following cardiac arrest due to ventricular arrhythmias. However, electrophysiological changes and proarrhythmic risk after amiodarone treatment have not yet been explored in TH.

### Methods

Epicardial high-density bi-ventricular mapping was performed in pigs under baseline temperature (BT), TH (32–34°C), and amiodarone treatment during TH. The total activation time (TAT), conduction velocity (CV), local electrogram (LE) duration, and wavefront propagation from pre-specified segments were analyzed during sinus rhythm (SR) or right ventricular (RV) pacing (RVP), along with tissue expression of connexin 43. The vulnerability to ventricular arrhythmias was assessed.

### Results

Compared to BT, TH increased the global TAT, decreased the CV, and generated heterogeneous electrical substrate during SR and RVP. During TH, the CV reduction and LE duration prolongation were greater in the anterior mid RV than in the other areas, which changed the wavefront propagation in all animals. Compared to TH alone, amiodarone treatment during TH further increased the TAT and LE duration and decreased the CV. Heterogeneous conduction was partially attenuated after amiodarone treatment. After TH and amiodarone treatment, the connexin 43 expression in the anterior mid RV was lower than that in the other areas, compatible with the heterogeneous CV reduction. The animals under TH and

**Funding:** This work was funded by the Ministry of Science and Technology of Taiwan (109-2314-B-075 -076 -MY3, 111-2314-B-075 -007 -MY3). Chin-Yu Lin is the recipient of the funding award. The funders had no role in study design, data collection and analysis, decision to publish, or preparation of the manuscript.

**Competing interests:** The authors have declared that no competing interests exist.

amiodarone treatment had a higher incidence of inducible ventricular arrhythmias than those under BT or TH without amiodarone.

## Conclusion

Electrical heterogeneity during amiodarone treatment and TH was associated with vulnerability to ventricular arrhythmias.

## Introduction

Therapeutic hypothermia (TH) improves the overall survival and neurological outcomes in patients resuscitated from cardiac arrest with ventricular arrhythmias [1–3]. However, it has been recently alerted that TH would increase the risk of arrhythmias [4]. TH increases the duration of ventricular depolarization and repolarization and decreases the conduction velocity (CV) through the modulation of ion channels, connexin 43 remodeling, and mitochondrial function [5–8]. These changes might contribute to the risk of pro-arrhythmia. Amiodarone is effective in suppressing ventricular arrhythmia and has therefore become the most frequently used antiarrhythmic drugs during TH following sudden cardiac arrest to prevent the recurrence of ventricular arrhythmias. However, the use of amiodarone also increases the duration of cardiac depolarization and repolarization by inhibiting sodium and potassium channels and decreasing the CV [9–11], which is proarrhythmic [12]. Although the cardiac electrophysiological effects of TH and amiodarone have been studied respectively [13, 14], the cardiac electrophysiological changes after amiodarone treatment during TH remain debatable.

Amiodarone treatment is associated with a low but could-not-be-ignored incidence of pro-arrhythmias [15]. Some researchers argued that the exacerbation of QT prolongation after the concurrent use of TH and amiodarone might potentiate the risk of ventricular arrhythmias [16]. By in vitro left ventricular (LV) transmural wedge preparation, amiodarone treatment during TH and ischemia increased the dispersion of repolarization and the CV, but not the incidence of conduction block or arrhythmia induction [4]. Although focal transmural heterogeneities were present in wedge preparation, global arrhythmia substrates in the intact heart were not addressed.

In this study, we aimed to evaluate cardiac electrophysiological changes in the entire heart after amiodarone treatment during TH and to determine the risk for ventricular arrhythmias after amiodarone treatment and TH in a pig model. In addition, the regional electrical heterogeneity after amiodarone treatment was comprehensively delineated via epicardial high-density mapping [17, 18]. The clarification of electrophysiological changes and the risk for pro-arrhythmias after the use of amiodarone during TH could critically guide future clinical decisions regarding anti-arrhythmic therapy in patients receiving TH.

## Methods

### Swine preparation

The details were described **S1 Table in S1 File**. The study protocol is summarized in **S1 Fig in S1 File.** In the step 1, six pigs underwent the study protocol I. Surface electrocardiography (ECG) was performed through four limb leads. A quadripolar steerable catheter was positioned in the right ventricle (RV) through femoral vein access to allow RV pacing (RVP). The location of the catheter was confirmed using echocardiography and fluoroscopy. A left

epicardial window was created through mini-thoracotomy, through which three-dimensional (3D) epicardial high-density mapping was performed. The 3D epicardial high-density maps in sinus rhythm (SR) or RVP were created sequentially while the pigs were kept under baseline temperature (BT, 36–37˚C), TH (32–34˚C, S1 File) [19], and amiodarone infusion during TH (amiodarone/TH). The ECG parameters, electrophysiological characteristics, and relevant 3D epicardial map parameters were compared and analyzed (S1 File). Amiodarone (10 mg/kg) was infused slowly for 20 min [20], and epicardial mapping was performed after the infusion under TH. Finally, the vulnerability to ventricular arrhythmias was assessed in three pigs via burst ventricular pacing, and the threshold for the induction of ventricular arrhythmias was determined. In the step 2, 4 and 4 swine underwent protocol II and protocol III respectively (**S1 Fig in S1 File**). The preparation of sheath, surface ECG, and epicardial window was the same as protocol I. Vulnerability to ventricular arrhythmia was performed in 4 and 4 swine after the induction of TH in protocol II and without induction of TH in protocol III respectively (**S1 Fig in S1 File**). External defibrillation was promptly performed when sustained ventricular arrhythmias occurred.

The authors confirmed that all experimental protocols were approved by Animal Experiment Committee of Taipei Veterans General Hospital.

The authors confirmed that all methods were carried out in accordance with thee Guide for the Care and Use of Laboratory Animals.

The authors confirmed that all methods are reported in accordance with ARRIVE guidelines

After completing the study protocol under general anesthesia, the pigs underwent intravenous injection of potassium chloride. Myocardial tissue from the RV or LV was collected immediately after the animals were sacrificed after intravenous injection of potassium chloride (1–2 mEq K+ /kg).

## ECG parameters and epicardial electrophysiological characteristics

The ECG parameters were measured and compared on the surface ECG lead II among BT, TH, and amiodarone/TH. The analyses included the QRS duration, QT interval, corrected QT (QTc) interval, and T-peak to T-end (TpTe) interval [21–23]. The 3D high-density mapping was performed via epicardial access using the Ensite Precision 2 and the Advisor™ HD Grid Mapping Catheter (Abbott, St. Paul, MN, USA), which has 16 electrodes with an equidistant electrode spacing of 3 mm. The average collection points for each 3D map were 5549 ± 4375. We set a reference of ECG morphology during sinus rhythm or RV pacing rhythm for creating a local activation time (LAT) map. We used the Score Map function of Ensite system to select identical QRS morphology as the reference setting. (sinus rhythm or RV pacing rhythm) The reference timing of LAT would be the onset of the QRS or the peak R wave of selected ECG lead. We assess LAT of electrogram based upon the peak of the bipolar electrogram by using the HD mapping catheter [24]. The time difference between the local electrogram and reference timing would be calculated and produced a LAT value [25]. The total activation time (TAT), bipolar voltage map, and activation map were collected from the global heart. The epicardial surface of the 3D geometry was further divided into 13 anatomical segments (**S2 Fig in S1 File** and next section). An isochronal map was applied to clarify the wavelet propagation on the global epicardium and each segment. The TAT, CV, bipolar voltage, and local electrogram (LE) duration were analyzed for each segment during SR and RVP. LE duration was measured to evaluate the characteristics of LE property [26].

Forty points were randomly selected from each pre-specified segment from each map of all pigs. The duration from the first sharp upstroke/deflection of the LE to the end of the LE from

the bipolar electrogram was the LE duration. The regional LE duration represented the mean value of the LE duration from randomly-selected 40 points from the pre-specified segment. The global LE duration is the average of mean LE durations from the 13-segment models. The global CV was measured by dividing the TAT by the anatomic distance from the earliest activated site to the latest activated site in the 3D geometry. The regional CV was measured by dividing the TAT within the pre-specified segment by the anatomic distance from the earliest activated site to the latest activated site within the pre-specified segment [25, 27].

### Definition of the anatomical segments in the 3D geometry

As the myocardial segments on a bull's-eye display shown in **S2 Fig in S1 File**, the epicardial myocardial segments were named and localized with reference to both the long axes of the ventricle and the 360-degree circumferential locations on the short-axis views, which were modified from the 17-segment model (American Heart Association) [28]. The basal, mid-cavity, and apical regions were used as part of the name to define the location along the long axis of the ventricle from the apex to the base. With regard to the circumferential location, the bases and mid-cavity slices were divided into six segments of 60° each. The circumferential locations in the basal and mid-cavity were anterior, RV anterior, RV posterior, posterior, and LV posterior. The left anterior descending artery (LAD) was used as an anatomical landmark for segmentation. The ostium of the LAD separated segments 1 and 6, while the distal LAD separated segments 7 and 8. Conversely, the ostium of the posterior interventricular artery (PIA) separated segments 3 and 4, while the distal PIA separated segments 11 and 10.

The method of Immunofluorescent staining, western blotting, and statistics were mentioned in the S1 File.

## Results

### Sequential changes in the surface ECG parameters, TAT, and voltage

The ECG parameters during SR were compared among BT, TH, and amiodarone/TH. The QRS durations (BT, TH, and amiodarone/TH: 99.1 ± 2.1, 106.5 ± 3.1, and 142.3 ± 4.6 ms, respectively), QT intervals (389.2 ± 9.4, 473.7 ± 8.5, and 536.7 ± 13.1 ms), QTc intervals (518.7 ± 11.9, 605.7 ± 14.7, and 659.7 ± 17.3 ms), and TpTe intervals (48.7 ± 1.0, 56.8 ± 2.0, and 86.7 ± 7.3 ms) all increased after TH and further significantly prolonged after amiodarone infusion (**Fig 1** and **S2 Table in S1 File**).

The global changes in the electrophysiological parameters are summarized in **Table 1**. After the induction of TH, the TATs during SR increased in comparison to those during BT (**Fig 2A**; BT vs. TH: 45.5 ± 3.4 vs. 51.2 ± 3.1 ms, respectively, P < 0.001). After amiodarone treatment, the TATs were prolonged (TH vs. Amiodarone/TH: 51.2 ± 3.1 vs. 53.3 ± 3.4 ms, respectively, P < 0.001). At either phase of BT, TH, or amiodarone/TH, the TATs during RVP were longer than those during SR (**Table 1**). During RVP, a similar sequential increase in the TATs after induction of TH or amiodarone/TH was still observed compared to the TATs during BT. Representative isochronal maps for BT, TH, and amiodarone/TH during SR are shown in **Fig 2B**. There were no differences in the mean bipolar voltage between the three phases during either SR or RVP (**Table 1** and **S3 Fig in S1 File**). No low-voltage zone (bipolar voltage: < 0.5 mV) was observed in any of the phases.

### Global and regional changes in the CV LE duration

A sequential decrease in the global CVs after TH and amiodarone/TH was observed during SR compared to the CVs during BT (**Fig 3**A and **Table 1**; BT, TH, and amiodarone/TH: 2.0 ± 0.1,

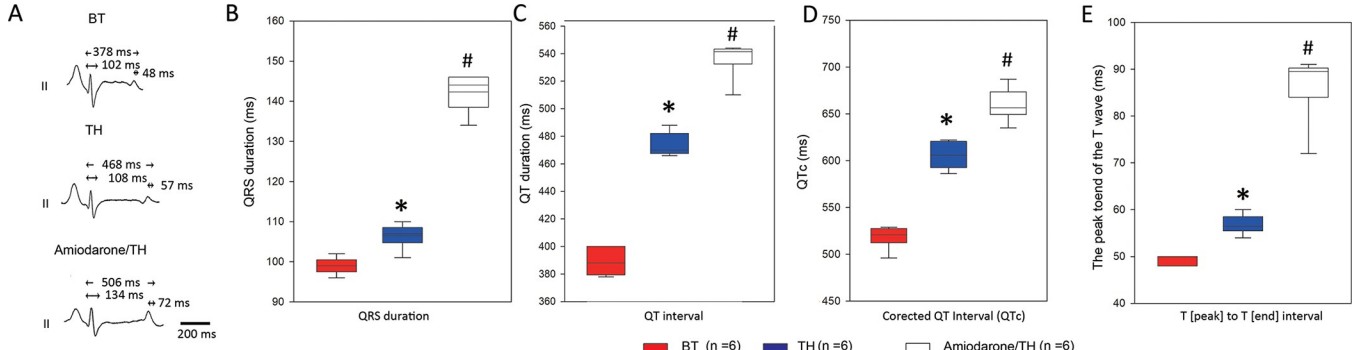

**Fig 1. Sequential changes in the surface ECG parameters after TH and amiodarone treatment.** (A) Representative ECG parameters (QRS duration, QT interval, and TpTe interval) at BT (upper panel), TH (middle panel), and amiodarone/TH (lower panel) of pig 3 during SR. The ECG parameters analyzed included the following: (B) QRS durations, (C) QT intervals, (D) QTc intervals, and (E) TpTe intervals. TH increased all ECG parameters compared with BT. However, even under TH, amiodarone treatment further increased the durations of all ECG parameters. These data were derived from all the six pigs from this study. *$P < 0.05$, BT vs. TH; #$P < 0.05$, TH vs. Amiodarone/TH, both by the paired t-test. The data were presented as means ± standard deviations. BT, baseline temperature; ECG, electrocardiography; QTc, corrected QT; TH, therapeutic hypothermia; TpTe, T-peak to T-end interval.

1.8 ± 0.1, and 1.76 ± 0.1 m/s, respectively). During RVP, a similar decrease in the CV was observed (**Table 1** and **Fig 3**A; 2.0 ± 0.1 vs. 1.8 ± 0.05 vs. 1.7 ± 0.1 m/s, respectively).

We further analyzed the CVs in the individual segments during SR and RVP (**Tables 2 and 3**). The CVs during BT were similar between the epicardial segments during SR and RVP; however, the CVs were significantly different between the segments during TH and amiodarone/TH. Therefore, the interval changes in the CVs between BT and TH (**Fig 3**B, ΔCV between BT and TH) and between TH and amiodarone/TH (ΔCV between TH and amiodarone/TH) were analyzed for each segment during SR and RVP, respectively. Although a reduction in the CVs was observed in all segments during TH and amiodarone/TH, the reduction in the ΔCV between BT and TH in the anterior mid RV (segment 8) was significantly greater than that in the other segments either during SR or RVP (**Fig 3**). After amiodarone treatment,

**Table 1. Electrophysiological parameters of global ventricular epicardium.**

|  | BT | HT | Amiodarone/HT |
|---|---|---|---|
| Total activation time (SR, ms) | 45.5±3.4 | 51.2±3.1[†] | 53.3±3.4[‡] |
| Total activation time (RVP, ms) | 81.5±4.5 | 86.0±5.2[†] | 88.3±5.7[‡] |
| *P-value*[*] | <0.001 | <0.001 | <0.001 |
| Bipolar Voltage (SR, ms) | 5.4±0.6 | 5.6±0.6 | 5.5±0.5 |
| Bipolar Voltage (RVP, ms) | 5.7±0.6 | 5.6±0.6 | 5.5±0.5 |
| *P-value*[*] | 0.112 | 0.720 | 0.974 |
| Conduction velocity (SR, m/s) | 2.0±0.1 | 1.8±0.1[†] | 1.76±0.07[‡] |
| Conduction velocity (RVP, m/s) | 2.0±0.1 | 1.8±0.05[†] | 1.7±0.1[‡] |
| *P-value*[*] | 0.368 | 0.320 | 0.210 |
| LE duration (SR, ms) | 42.5±4.2 | 47.9±5.8[†] | 49.9±6.1[‡] |
| LE duration (RVP, ms) | 43.7±6.0 | 48.1±7.5[†] | 49.5±7.4[‡] |
| *P-value*[*] | 0.142 | 0.815 | 0.866 |

[*] SR vs. RVP

[†] P<0.05, BT vs. TH

[‡] P <0.05, TH vs. amiodarone/TH, all the analyses by the paired t test

BT: baseline temperature; HT: therapeutic hypothermia; LE: local bipolar electrograms; SR: sinus rhythm; RVP: right ventricular pacing.

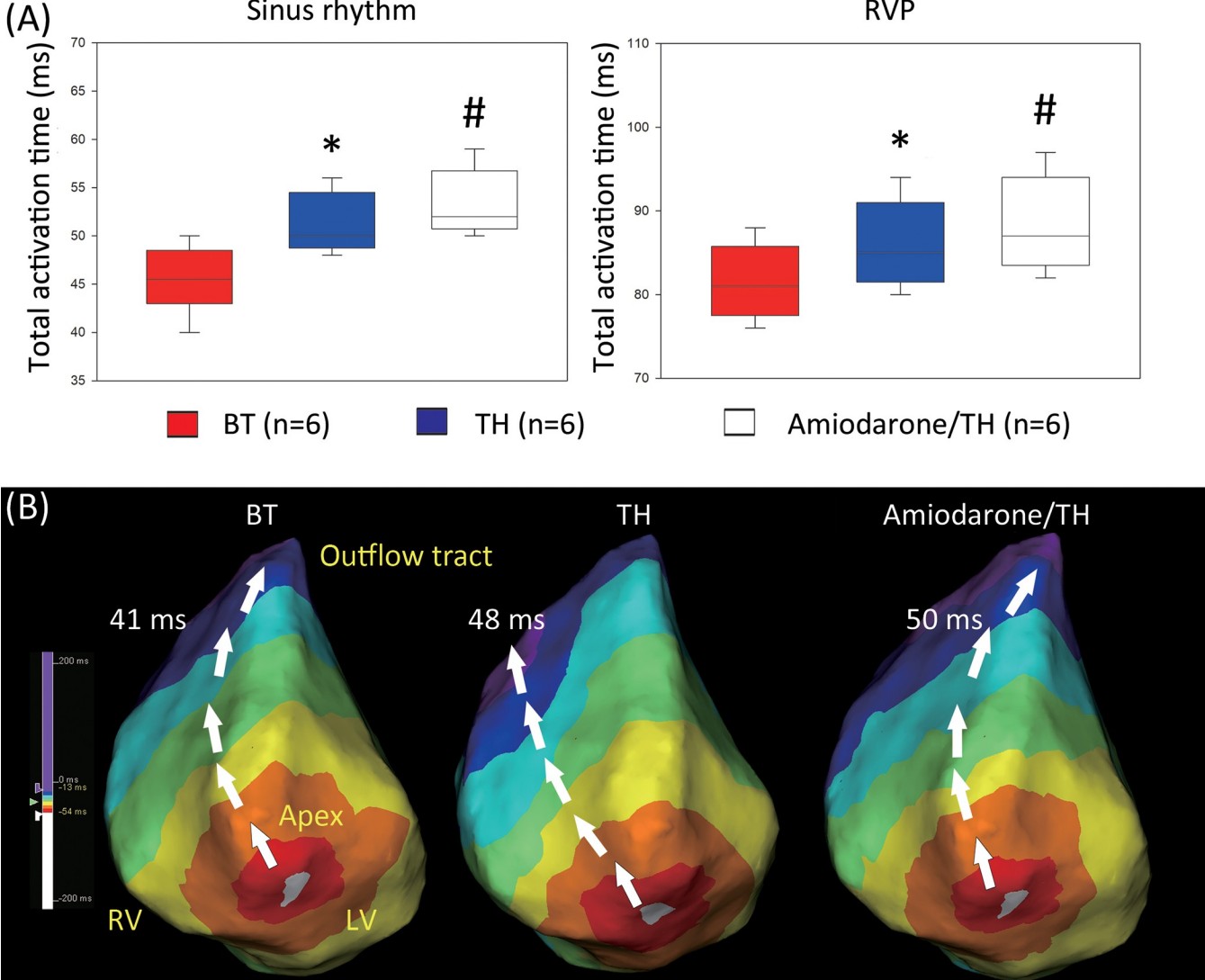

**Fig 2. Sequential changes in the TAT after TH and amiodarone treatment.** (A) Sequential changes in the TAT after the induction of TH and amiodarone infusion (amiodarone/TH) either during SR (left panel) or RVP (right panel). These data were derived from the six pigs from protocol I. (B) Representative isochronal map of pig 1 during SR (left panel: BT, middle panel: after TH, and right panel: after amiodarone infusion). The Panel B showed a left anterior-oblique (LAO) view to the heart. The upper part indicated anterior aspect of the heart and the lower part indicated the posterior aspect of the heart. From a LAO view, the LV was in the right side and RV in the left side. The landmark was highlighted in yellow color. The white color on the map indicated the earliest activation site and followed by red, orange, yellow, green, light blue, blue, and purple color. *P < 0.05, BT vs. TH; #P < 0.05, TH vs. Amiodarone/TH, both by the paired t-test. The data are presented as means ± standard deviations. BT, baseline temperature; LV, left ventricle; RV, right ventricle; RVP, right ventricular pacing; SR, sinus rhythm; TAT, total activation time; TH, therapeutic hypothermia.

the ΔCV between TH and amiodarone/TH in segment 8 was marginally lower than that in the other segments (segment 8 vs. other segments: -0.04 ± 0.03 vs. -0.09 ± 0.07 m/s, respectively, P = 0.07) during SR. However, no significant inter-segmental difference in the ΔCV between TH and amiodarone/TH was observed during RVP.

A sequential prolongation of the global LE duration was observed during SR (**Table 2** and **Fig 4**A; BT, TH, and amiodarone/TH: 42.5 ± 4.2 vs. 47.9 ± 5.8 vs. 49.9 ± 6.1 ms, respectively), as well as during RVP. The LE durations during BT were similar between the epicardial segments during SR and RVP; however, they significantly differed between the segments during

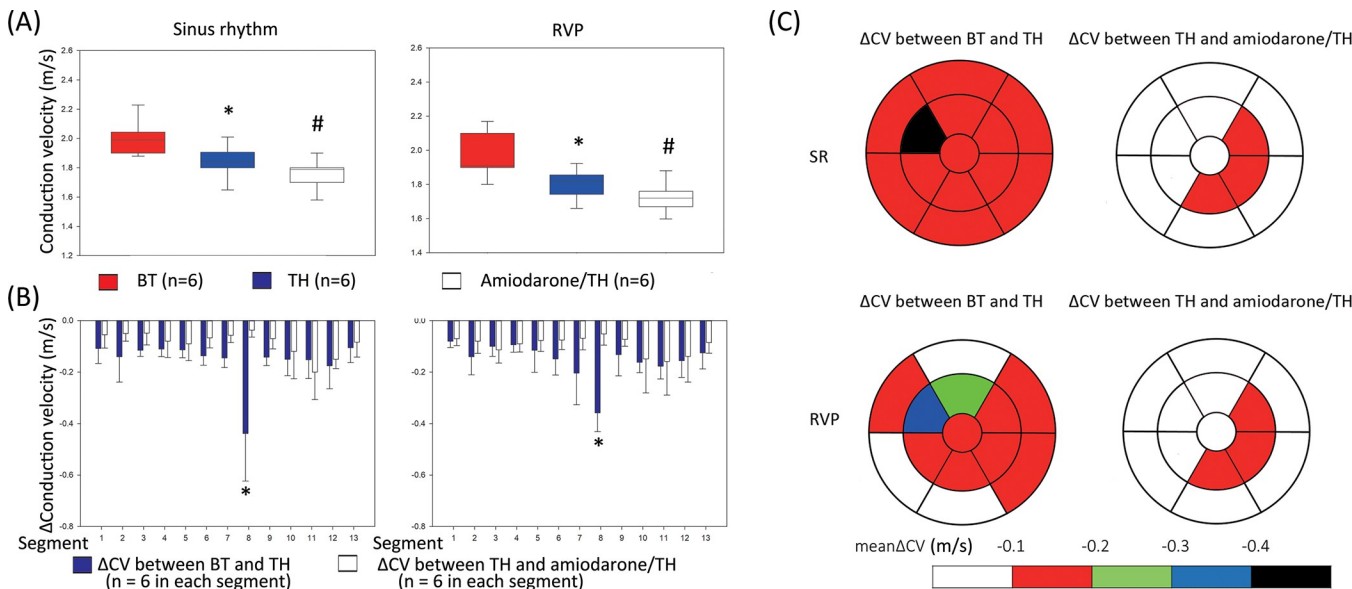

**Fig 3. Global and regional changes in the CV.** (A) Sequential changes in the CV after TH or amiodarone treatment during SR or RVP. A sequential reduction in the CVs was observed after TH and amiodarone treatment. These data were derived from all the six pigs from this study. *P < 0.05, BT vs. TH; #P < 0.05, TH vs. Amiodarone/TH, both by the paired t-test. (B) Interval changes between BT and TH (ΔCV between BT and TH) and between TH and amiodarone/TH (ΔCV between TH and amiodarone/TH) in the different epicardial segments. These data were derived from all the six pigs from this study. The interval changes in the CV in segment 8 (anterior mid right ventricle) were significantly greater than those in all the other segments. *P < 0.05, ΔCV between BT and TH in segment 8 vs. the other segments in the post-hoc analysis with the Bonferroni method. (C) The color-coded polar map of the interval changes between BT and TH (ΔCV between BT and TH) and between TH and amiodarone/TH (ΔCV between TH and amiodarone/TH) in the different epicardial segments. Each color indicated different degree of decrease in the CV.

**Table 2. Electrophysiological parameters in different epicardial segments during sinus rhythm.**

| Segment | Durations of local electrogram (LE, ms) | | | Conduction velocity (CV, m/s) | | |
|---|---|---|---|---|---|---|
| | BT | TH | Amiodarone/TH | BT | TH | Amiodarone/TH |
| 1 | 42.1±3.8 | 44.8±3.3 | 47.8±4.5 | 1.9±0.1 | 1.8±0.2 | 1.7±0.1 |
| 2 | 42.7±4.4 | 47.5±4.3 | 49.2±5.5 | 2.0±0.1 | 1.8±0.1 | 1.8±0.1 |
| 3 | 43.3±5.1 | 47.1±5.3 | 48.7±5.8 | 1.9±0.1 | 1.8±0.1 | 1.7±0.1 |
| 4 | 42.9±3.6 | 45.4±2.6 | 47.1±3.1 | 2.0±0.2 | 1.9±0.2 | 1.9±0.2 |
| 5 | 41.7±4.6 | 45.5±3.4 | 47.9±4.4 | 2.1±0.2 | 1.9±0.2 | 1.8±0.2 |
| 6 | 43.8±5.0 | 48.2±4.5 | 50.4±5.3 | 2.0±0.2 | 1.9±0.2 | 1.8±0.2 |
| 7 | 41.8±4.2 | 45.7±2.7 | 48.2±3.6 | 2.0±0.2 | 1.9±0.2 | 1.8±0.2 |
| 8 | 43.8±5.2 | 60.9±5.3 | 61.6±5.8 | 2.0±0.1 | 1.5±0.2 | 1.5±0.2 |
| 9 | 45.5±5.2 | 53.9±6.2 | 55.3±7.0 | 1.9±0.1 | 1.7±0.1 | 1.6±0.1 |
| 10 | 43.0±3.2 | 45.4±2.7 | 48.1±5.1 | 2.0±0.2 | 1.9±0.1 | 1.8±0.04 |
| 11 | 39.8±3.7 | 45.5±4.7 | 47.8±4.5 | 2.1±0.2 | 2.0±0.1 | 1.8±0.04 |
| 12 | 40.5±4.1 | 45.3±3.7 | 48.2±5.0 | 2.1±0.2 | 2.0±0.1 | 1.8±0.1 |
| 13 | 41.4±3.5 | 46.9±3.1 | 49.0±5.1 | 2.00±0.1 | 1.9±0.1 | 1.8±0.1 |
| *P-value** | 0.717 | <0.001 | <0.001 | 0.016 | <0.001 | <0.001 |

*P- value between segment 1 to 13 by one-way ANOVA.

The LE was significantly prolonged after TH (BT vs. TH) and after the infusion of amiodarone (TH vs. Amiodarone/TH) in all segments respectively.

The CV was significantly decreased after TH (BT vs. TH) and after the infusion of amiodarone (TH vs. Amiodarone/TH) in all segments respectively.

**Table 3. Electrophysiological parameters in different epicardial segments during right ventricular pacing.**

| Segment | Durations of local electrogram (ms) | | | Conduction velocity (m/s) | | |
|---|---|---|---|---|---|---|
| | BT | TH | Amiodarone/TH | BT | TH | Amiodarone/TH |
| 1 | 43.7±3.9 | 44.6±3.9 | 47.0±3.5 | 1.9±0.1 | 1.8±0.1 | 1.7±0.1 |
| 2 | 44.5±5.7 | 47.2±6.4 | 49.1±5.8 | 1.9±0.1 | 1.8±0.1 | 1.7±0.1 |
| 3 | 43.0±8.1 | 45.2±8.0 | 46.6±7.0 | 1.9±0.1 | 1.8±0.1 | 1.7±0.1 |
| 4 | 41.6±6.9 | 45.3±8.3 | 47.1±8.3 | 1.9±0.1 | 1.8±0.1 | 1.7±0.1 |
| 5 | 41.4±4.8 | 44.5±5.1 | 46.3±4.2 | 1.9±0.2 | 1.8±0.1 | 1.7±0.1 |
| 6 | 41.5±8.1 | 45.1±7.7 | 48.3±3.6 | 1.9±0.1 | 1.8±0.1 | 1.7±0.1 |
| 7 | 41.8±10.0 | 45.5±9.9 | 46.8±9.6 | 1.9±0.1 | 1.7±0.1 | 1.7±0.1 |
| 8 | 45.2±5.8 | 58.1±8.4 | 59.4±9.1 | 1.9±0.1 | 1.6±0.04 | 1.5±0.1 |
| 9 | 45.3±8.9 | 56.4±8.5 | 57.4±8.5 | 1.9±0.1 | 1.8±0.1 | 1.7±0.1 |
| 10 | 45.6±3.0 | 49.0±4.2 | 49.9±4.6 | 2.1±0.2 | 1.9±0.2 | 1.8±0.1 |
| 11 | 44.9±4.5 | 48.0±4.8 | 49.2±4.0 | 2.1±0.2 | 1.9±0.1 | 1.8±0.1 |
| 12 | 44.0±2.9 | 47.8±3.9 | 49.6±4.1 | 2.1±0.1 | 1.9±0.1 | 1.8±0.1 |
| 13 | 45.7±3.1 | 48.8±4.9 | 50.1±4.1 | 2.0±0.1 | 1.8±0.04 | 1.8±0.1 |
| P-value* | 0.947 | 0.009 | 0.008 | 0.019 | <0.001 | <0.001 |

*P- value between segment 1 to 13 by one-way ANOVA.

The LE was significantly prolonged after TH (BT vs. TH) and after the infusion of amiodarone (TH vs. Amiodarone/TH) respectively.

The CV was significantly decreased after TH (BT vs. TH) and after the infusion of amiodarone (TH vs. Amiodarone/TH) respectively.

TH and amiodarone/TH (**Tables 2, 3**). The interval changes in the LE durations between BT and TH (**Fig 4B**, ΔLE between BT and TH) and between TH and amiodarone/TH (ΔLE between TH and amiodarone/TH) were analyzed for each segment during SR and RVP, respectively.

Although a prolongation of the LE duration was observed in all segments during TH and amiodarone/TH, the increase in the ΔLE between BT and TH in the anterior mid RV (segment 8) and posterior mid RV (segment 9) was greater than that in the other segments either during SR or RVP (**Fig 4B**). After amiodarone treatment, the increase in the ΔLE between TH and amiodarone/TH in segment 8 or 9 was marginally smaller than that in the other segments (segment 8 vs. segment 9 vs. other segments: 0.7 ± 1.7 vs. 1.4 ± 1.4 vs. 2.3 ± 2.2 ms, respectively, P = 0.145) during SR. No significant inter-segmental difference in the ΔLE between TH and amiodarone/TH was observed during RVP. Representative recordings of the LE during SR at BT, TH, and amiodarone/TH are shown in **Fig 4C**.

## Changed wavelet activation pattern, vulnerability to ventricular arrhythmias, and connexin 43 expression

The inter-segmental wavelet propagation was analyzed using the isochronal map with pre-specified segments during SR and RVP (**Fig 5**, **Table 4**, and **S4 Fig in S1 File**). During SR, the latest activated segment was the outflow tract (segment 1) at BT for all pigs. After TH, the latest activated segments were all changed to the anterior basal RV (segment 2) in all pigs. After the treatment of amiodarone under TH, the latest activated segment returned to the outflow tract (segment 1) in all six pigs. The latest activated site during TH was different from that during BT and amiodarone/TH (P < 0.001); however, those between BT and amiodarone/TH were the same. During RVP, the latest activated segment was mainly the posterior basal LV (segment 5) in five pigs (83.3%) at BT. After TH, the latest activated site changed to the anterior basal LV (segment 6, four pigs [66.7%]). Amiodarone treatment did not restore the activation

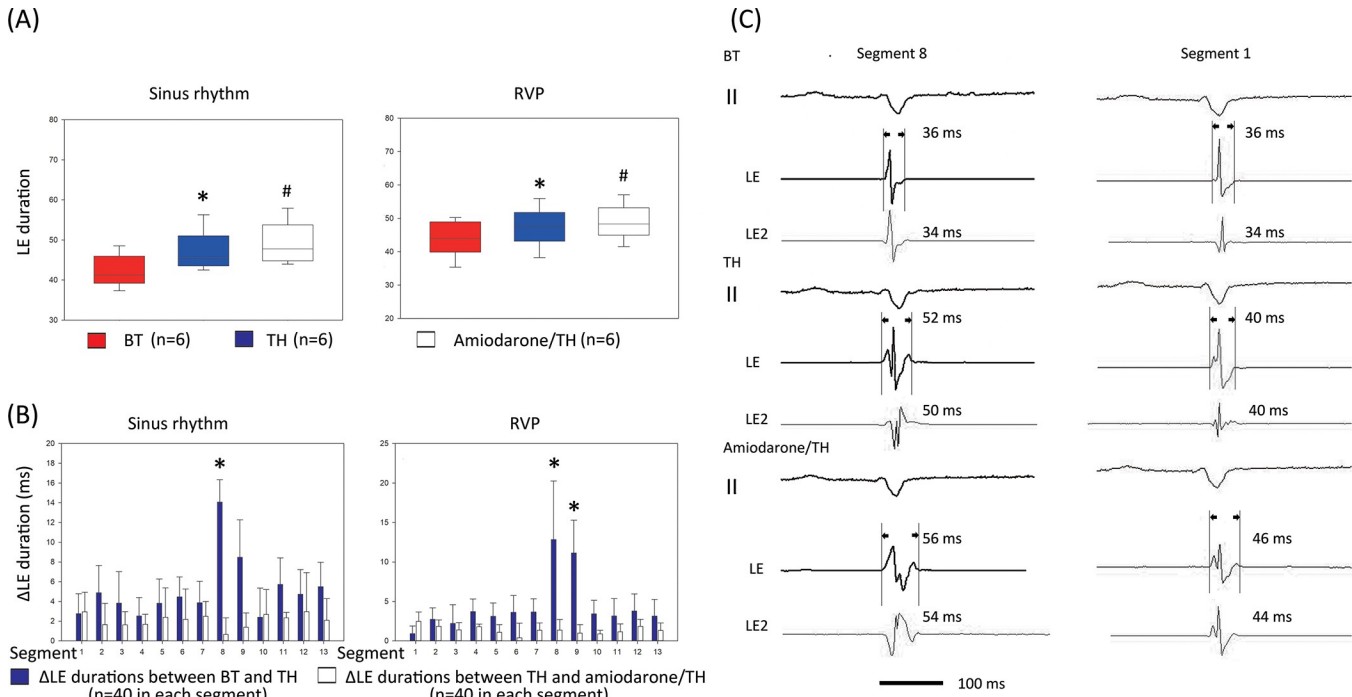

**Fig 4. Global and regional changes in the LE duration.** (A) Sequential changes in the LE duration after TH or amiodarone treatment during SR or RVP. These data were derived from all the six pigs from this study. A sequential reduction in the LE durations was observed after TH or amiodarone treatment. *P < 0.05, BT vs. TH; #P < 0.05, TH vs. Amiodarone/TH, both by the paired t-test. (B) Interval changes between BT and TH (ΔLE durations between BT and TH) and between TH and amiodarone/TH (ΔLE durations between TH and amiodarone/TH) in the different epicardial segments. These data were derived from all the six pigs from this study. The interval changes in the LE durations in segment 8 (anterior mid RV or posterior mid RV) were significantly greater than those in all the other segments. *P < 0.05, ΔLE durations between BT and TH in segment 8 or segment 9 vs. the other segments in the post-hoc analysis with the Bonferroni method. (C) Representative bipolar electrograms of segment 8 and segment 1 from pig 4 during BT, TH, and amiodarone/TH demonstrating sequential prolongation of the LE durations.

pattern of BT. At amiodarone/TH, the posterior basal LV (segment 5) remained as the latest activated site in three pigs (50.0%). There was no significant changes in the latest activation patterns between BT, TH, and amiodarone/TH (P = 0.207). Representative patterns of the wavelet propagation are shown in **Fig 5A** and **S4 Fig in S1 File**. Sustained VT/VF was induced in all pigs (3/3, 100%) under TH and amiodarone treatment, in none of the pigs (0/4, 0%) under TH only, and in one of the control pigs (1/4, 25%). (**Fig 5**B) The vulnerability to ventricular arrhythmias of pigs in the amiodarone/TH group was higher than those at BT or TH. The threshold for inducing ventricular arrhythmias in the animals under amiodarone/TH was 270, 280, and 300 ms, respectively (283.3 ± 15.3 ms). In the only control animal with inducible ventricular arrhythmias, the threshold was 290 ms.

The expression of connexin 43 was not different between the controls and study group (**S5 Fig in S1 File**). As the CV was significantly different between segment 8 (anterior mid RV) and the other segments, we compared the connexin 43 expression in segment 8 to that in two other segments (segment 5: posterior basal LV and segment 12: anterior mid LV) via immunofluorescent staining. The connexin 43 expressions were lower in the anterior mid RV (**Fig 6A**, segment 8: 12.3 ± 4.8 a.u.) than in the LV (segment 5: 14.5 ± 4.7 a.u.; segment 12: 15.6 ± 6.9 a.u., P = 0.027). The connexin 43 lateralization was similar in different segments (**Fig 6B**, segment 5: 21.3 ± 5.2%; segment 8: 21.4 ± 6.8%; segment 12: 21.1 ± 6.7%, P = 0.946).

There was no significant difference of connexin 43 expression between the control group and the study group (**S5 Fig in S1 File**).

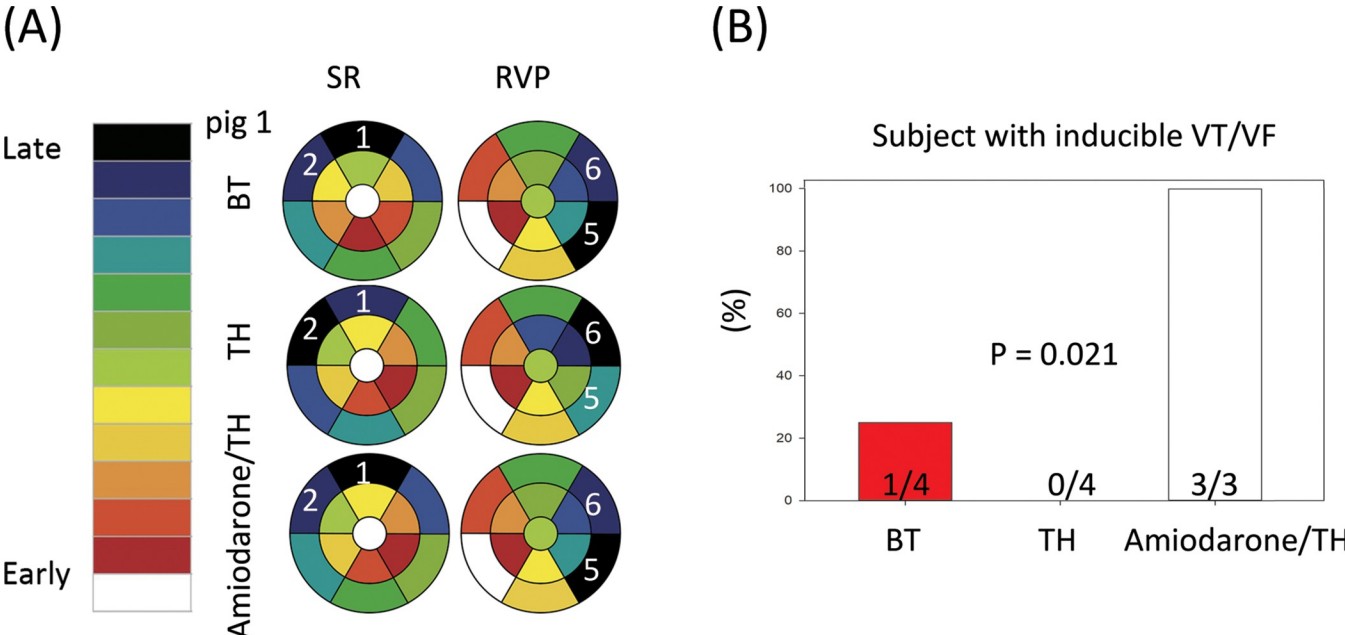

**Fig 5. Changes in the wavelet activation pattern, the vulnerability to ventricular arrhythmias after amiodarone treatment during TH, and Regional differences in the connexin 43 expression during TH and amiodarone treatment.** (A) Representative isochronal activation from pig 1 according to the pre-specified segments. During SR, the earliest activated site was the apical area (segment 13), and the latest activated site was the outflow tract area (segment 1) at BT. After TH, the latest activated site changed to the anterior basal RV (segment 2). After amiodarone infusion during TH, the latest activated site moved back to the outflow tract area (segment 1). During RVP, the earliest activated site was the posterior basal right ventricle (segment 3), and the latest activated site was mostly (83.3%) the posterior basal LV (segment 5) at BT. At TH, the latest activated site mostly changed to the anterior basal LV (segment 6, 66.7%). At amiodarone/TH, the latest activated site moved to the posterior basal LV (segment 5, 50%) in half of the pigs. (B) The incidence of ventricular arrhythmias after burst RVP. The vulnerability to ventricular arrhythmias was tested in 4, 4, and 3 pigs with BT, TH, and amiodarone/TH respectively. The vulnerability to ventricular arrhythmias of pigs with amiodarone/TH group was higher than the pigs at BT and the pigs with TH (P = 0.021).

## Discussion

The combination of amiodarone treatment and TH made the myocardium more vulnerable to arrhythmias induced by burst ventricular pacing. TH induced regional heterogeneity in the CV and LE duration in the anterior and posterior mid RVs; therefore, the patterns of wavelet propagation of the entire heart were altered. After amiodarone treatment during TH, amiodarone further increased QT prolongation, decreased the global CV, and increased the activation time. The patterns of wavelet propagation of the entire heart remained significantly altered during RVP after amiodarone treatment during TH. All these factors contributed to the

**Table 4. The latest epicardial activation site after hypothermia or the treatment of amiodarone.**

|  | BT | TH | Amiodarone/ TH | P-value* |
|---|---|---|---|---|
| **Latest activation zone in SR** |  |  |  |  |
| Segment 1 (outflow tract) | 6 (100.0%) | 0 (0.0%) | 6 (100.0%) | < 0.001 |
| Others than segment 1 | 0 (0.0%) | 6 (100.0%) | 0 (0.0%) |  |
| **Latest activation zone in RVP** |  |  |  |  |
| Segment 5 (posterior basal LV) | 5 (83.3%) | 2 (33.3%) | 3 (50.0%) | 0.207 |
| Others than segment 5 | 1 (16.7%) | 4 (66.7%) | 3 (50.0%) |  |

BT: baseline temperature; HT = therapeutic hypothermia; SR: sinus rhythm; LV: left ventricle; RVP: right ventricular pacing.

*P-value by the Chi-Square test

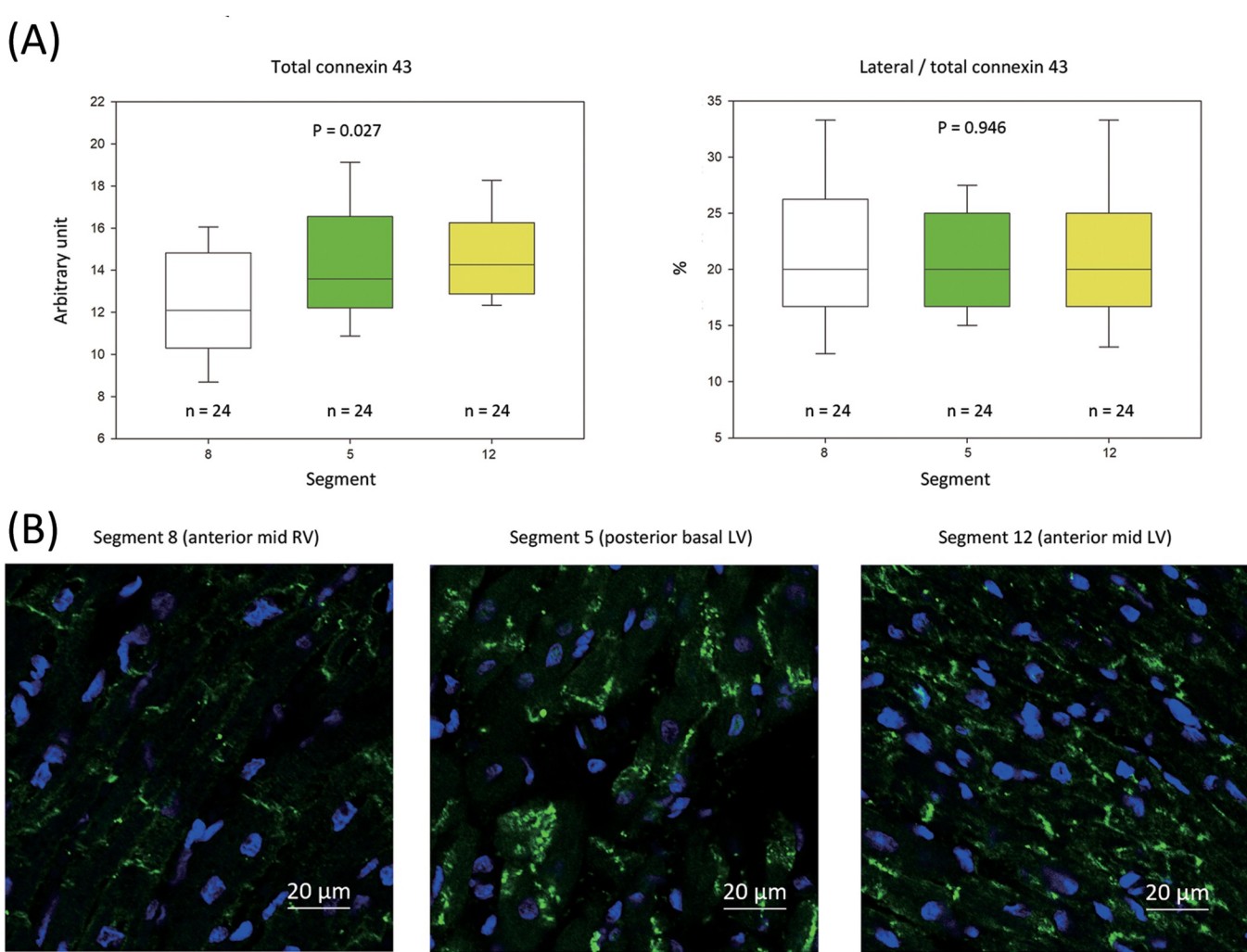

**Fig 6. Regional differences in the connexin 43 expression during TH and amiodarone treatment.** (A) Left panel: Comparison of the connexin 43 expressions among segment 8 (anterior mid RV wall), segment 5 (posterior basal LV wall), and segment 12 (anterior mid LV wall). The data are presented as box plot. These data were derived from pig 3, pig 4, pig 5, and pig 6 from the protocol I. Right panel: Comparison of the connexin 43 lateralization among segment 8 (anterior mid RV wall), segment 5 (posterior basal LV wall), and segment 12 (anterior mid LV wall). The data are presented as box plot. These data were derived from pig 3, pig 4, pig 5, and pig 6 from the protocol I. (6 slides from each segment of each pig) The parameters between the three segments were compared using one-way ANOVA. (B) Presentative fluorescent images of the connexin 43 expressions. Left panel: anterior mid RV (segment 8 of pig 6); middle panel: posterior basal LV (segment 5 of pig 5); right panel: anterior mid LV (segment 12 of pig 4). Green fluorescence: connexin 43; blue fluorescence: 4,6-diamidino-2-phenylindole for nucleus. BT: baseline temperature; LV = left ventricle; RV = right ventricle; RVP = right ventricular pacing; TH = therapeutic hypothermia.

increased incidence of inducible ventricular arrhythmias after amiodarone treatment during TH.

While the dispersion of repolarization and number of ventricular ectopies were either unchanged or decreased, TH increased the risk for cardiac alternans and ventricular arrhythmias during cardiac reperfusion in a porcine translational model of resuscitation from ischemic cardiac arrest [29]. Amiodarone treatment further increased the dispersion of repolarization and CV during TH [4]. Rather than ex vivo experiments on small tissue wedges, our work further provided information on the dynamic change in the global and regional electrical substrates in pig hearts after TH and amiodarone treatment. TH slowed down cardiac conduction as indicated by the changes in the TAT, CV, and LE duration. In addition, it

increased the regional heterogeneity, a vulnerable electric substrate for ventricular arrhythmias. The recent study demonstrated more hemodynamic-compromised ventricular arrhythmia could be observed in the patient received TH than in the patient without TH [4]. This is compatible with substrate change after TH in our present study.

Amiodarone treatment further worsened the reduction in global cardiac conduction and only partially reversed the regional heterogeneity in our study. The wavelet propagation pattern during amiodarone/TH during ectopic ventricular firing, as presented by RVP, remained similar to that after TH. Additionally, amiodarone infusion further increased QT, QTc, and TpTe intervals under TH, which was associated with an increased risk of ventricular arrhythmia [30]. The incidence of ventricular arrhythmias during amiodarone/TH was higher than that in BT and TH. These findings suggest that amiodarone might not be used without considering it potential risk of pro-arrhythmia. The present result was different from a previous report suggesting that amiodarone treatment was not associated with a different incidence of ventricular arrhythmias [4]. A previous study has used an in vivo pig model of resuscitation and analyzed spontaneous VF during resuscitation [4]. Although amiodarone treatment was not associated with a different incidence of ventricular arrhythmias, previous studies were probably underpowered as the number of pig experiments was only three [4]. In addition, the difference between our findings and previous findings can also be explained by the experimental designs. Our study did not induce coronary artery occlusion and performed resuscitation before TH and amiodarone treatment. Although our present study suggested amiodarone combined with TH might be associated with increased ventricular arrhythmia, the clinical evidence remains controversial and the present findings should be applied with caution in clinical practice.

The impact of amiodarone alone without TH was not investigated in our present study. Studies on amiodarone have revealed that intravenous amiodarone has minimal effect on the refractory periods of the ventricular muscle in baseline condition [31, 32] and might decrease conduction velocity in the ventricle [33, 34] A recent study by Piktel et al. [4] showed that amiodarone did not affect CV, dispersion of repolarization, or action potential duration during ischemia at 36˚C. Our study echoed the finding of the decrease in the CV after amiodarone infusion even under the TH.

Cardiac CV is determined by three factors [35]: (i) membrane excitability, controlled by voltage-gated sodium channels; (ii) electrical cell-to-cell coupling, mediated by gap junction proteins (connexins), and (iii) cardiac tissue fibrosis. CV was decreased heterogeneously in our present study. The voltage mapping didn't reveal low-voltage area and the expression of connexin 43 was accessed. Regional differences in the pan-connexin 43 expression were observed in the animals receiving TH and amiodarone treatment, as indicated by the dominant reduction in the pan-connexin 43 expression levels in the anterior mid RV wall. The reduction in the pan-connexin 43 expression levels is an early event that precedes CV slowing [36]. The highest reduction in the CV was observed over the anterior RV wall, which was compatible with the area with the lowest connexin 43 expression. Therefore, the reduction in the pan-connexin 43 expression levels may partially explain the potential mechanism of the heterogenous electrical substrates after TH and amiodarone treatment. Inhomogeneous reduction in the non-phosphorylated connexin 43 expression levels were observed in the anterior RV wall compared to that in other areas in rats receiving a short duration of TH of 30 min [7]. This finding is compatible with that of our work.

The inter-segmental wavelet propagation was determinate by CV of each segment. In our present study, we investigated the epicardial CV and electrogram of each segment. We observed a significant reduction in the CV after induction of TH in the anterior mid RV (segment 8), which was greater than that in the other segments, which caused the wavelet

propagation change from BT to TH. After the infusion of amiodarone under TH, the CV of each segment was further decreased and alleviated the difference between the anterior mid RV (segment 8) and other segments. This change in CV resulted in the reverse of inter-segmental wavelet propagation inter-segmental wavelet propagation. The mechanism might be related to the change of distribution of connexin 43 or the heterogenous change of CV in the endocardium/transmural myocyte [37]. The amiodarone could provide heterogeneous effect on the myocardium and decrease the ventricular repolarization dispersion [38]. Our study indicated the amiodarone might provide heterogenous effect on the myocardial CV under TH. Further study investigating the electrophysiological parameter with simultaneous endocardial and epicardial mapping to clarify the is warrant.

## Limitations

First, the experiments were performed in healthy subjects, which could be different from those with cardiac arrest in the clinical scenario. Second, the ventricle was mapped using only epicardial access and complete endo-epicardial ventricular electrophysiological information was not available in this study. Third, the Ventricular arrhythmia could also be attributed to abnormal ventricular repolarization [39, 40]. In our study, we only measured the parameters as QT interval, QTc interval, and TpTe on surface ECG to identify repolarization abnormality. The value of TpTe in our present study might only reflect global dispersion but not transmural dispersion, and its usefulness as a marker of arrhythmia has been questioned [41]. The system of high-density mapping in the present study could not measure cardiac repolarization. Therefore, future studies with optical mapping or monophasic action potential mapping are needed to identify regional repolarization and its link with arrhythmic attacks. Fourth, the measurements of TpTe, QRS duration, and QT interval were derived from the six limb leads of surface ECG rather than a standard 12-lead ECG. Fifth, the number in animals for vulnerability test for ventricular arrhythmias is small for a solid conclusion.

## Conclusions

Electrical heterogeneity during amiodarone treatment and TH was associated with vulnerability to ventricular arrhythmias.

## Supporting information

**S1 File.**
(DOCX)

## Author Contributions

**Conceptualization:** Chin-Yu Lin, Yu-Feng Hu, Shih-Ann Chen.

**Data curation:** Chin-Yu Lin, Yu-Feng Hu.

**Formal analysis:** Chin-Yu Lin, Yu-Feng Hu.

**Funding acquisition:** Chin-Yu Lin, Yu-Feng Hu, Shih-Ann Chen.

**Investigation:** Chin-Yu Lin, Yu-Feng Hu, Shih-Ann Chen.

**Methodology:** Chin-Yu Lin, Yu-Feng Hu, Shih-Ann Chen.

**Project administration:** Chin-Yu Lin, Yu-Feng Hu, Shih-Ann Chen.

**Resources:** Chin-Yu Lin, Yu-Feng Hu, Shih-Lin Chang, Shih-Ann Chen.

**Software:** Chin-Yu Lin, Yu-Feng Hu.

**Supervision:** Yu-Feng Hu, Yi-Jen Chen, Shih-Lin Chang, Li-Wei Lo, Ta-Chuan Tuan, Shih-Ann Chen.

**Validation:** Yu-Feng Hu.

**Visualization:** Yu-Feng Hu, Hung-I Yeh.

**Writing – original draft:** Chin-Yu Lin.

**Writing – review & editing:** Ting-Yung Chang, Yu-Feng Hu, Yu-Cheng Hsieh, Yi-Jen Chen, Hung-I Yeh, Yenn-Jiang Lin, Shih-Lin Chang, Li-Wei Lo, Tze-Fan Chao, Fa-Po Chung, Jo-Nan Liao, Shih-Ann Chen.

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
