## [Decision Letter · Decision Letter 0]

19 Dec 2022

PONE-D-22-27467Epicardial electrical heterogeneity after amiodarone treatment increases vulnerability to ventricular arrhythmias under therapeutic hypothermiaPLOS ONE

Dear Dr. HU,

Thank you for submitting your manuscript to PLOS ONE. Your study was evaluated by an expert in the field and myself. The reviewer is concerned with several significant points regarding study methods and designs. Please read the comments carefully and address the issues accordingly.  

We look forward to receiving your revised manuscript.

Kind regards,

Tomohiko Ai, M.D., Ph.D.

Academic Editor

PLOS ONE

Journal Requirements:

“Ministry of Science and Technology of Taiwan (109-2314-B-075 -076 -MY3, MOST 110-2314-B-075 -063 -MY3), Grant of TVGH (V109B-013, V109C-005, V110C-039; V110B-043).”

Reviewers' comments:

Reviewer's Responses to Questions

**Comments to the Author**

1. Is the manuscript technically sound, and do the data support the conclusions?

Reviewer #1: Partly

2. Has the statistical analysis been performed appropriately and rigorously? 

Reviewer #1: No

3. Have the authors made all data underlying the findings in their manuscript fully available?

Reviewer #1: Yes

4. Is the manuscript presented in an intelligible fashion and written in standard English?

Reviewer #1: No

5. Review Comments to the Author

Reviewer #1: Using swine, authors　examined　the　electrophysiological　changes　and　proarrhythmic　risk　due　to　amiodarone　administration in therapeutic hypothermia (TH) and concluded that amiodaron and TH was associated with vulnerability to ventricular arrhythmias. The study is important in the use of amiodarone for the ventricular arrhythmias while TH, however, there are several comments to this study.

1. Why didn’t you check the effect of amiodaron at baseline temperature (BT)? Previous study by Piktel, et al. showed that amiodarone did not affect CV, DOR, or action potential duration during ischemia at 36°C. As you know, amiodaron changes various electrophysiological properties in the heart. If you could not examine the effect of amiodaron at BT by yourself, please cite adequate reference to compare the effect of amiodaron in BT and hypothermia.

2. You prepared 6 pigs for protocol I, however, there are only data from 3 pigs for the evaluation of vulnerability to ventricular arrhythmias. Why did you discard the data from other 3 pigs?

3. I could not find the detail of protocol II and III. In protocol I, you performed burst right ventricular pacing after amiodaron injection in TH. In protocol II, did you perform burst pacing just after TH, and in protocol III, you just performed burst pacing? Did you measure the baseline characteristics of 4 and 4 pigs in protocol II and III? Were the baseline electrophysiological characteristics of pigs in protocol II and III same with those in protocol I? Please revise the supplemental figure 1B and figure legend for the easy understanding of the readers.

4. Why did the inter-segmental wavelet propagation change from BT to TH and return to the same pattern with BT after amiodaron/TH? Please discuss the mechanism in the discussion section.

5. How did you obtain the cardiomyocytes for Immunofluorescent staining and western blotting, biopsy or after sacrifice? How did you treated the heart if you obtain the cardiomyocytes after sacrifice? Did you obtain the cardiomyocytes from endomyocardium or epicardium?

6. Please describe how to calculate the total and lateralization of connexin43 using image J software.

7. The sample numbers in graphs in figure 6A is 24. Did you obtain 6 specimens from each pig? Please add the explanation in the methods.

8. Did you check the connexin 43 levels in pigs assigned to protocol II and III? Conduction velocity was changed after TH or TH/amiodaron therapy. Showing connexin 43 levels in the BT in each segment are important to explain the relationship between conduction velocity change and the expression level of connexin 43.

8. In the cardiac conduction system, connexin 45 is highly expressed. Did you evaluate the expression level of connexin 45?

6. PLOS authors have the option to publish the peer review history of their article (what does this mean?). If published, this will include your full peer review and any attached files.

Reviewer #1: No

---

## [Author Response · Author response to Decision Letter 0]

16 Jan 2023

Responses to Reviewer 1

# 1. Why didn’t you check the effect of amiodaron at baseline temperature (BT)? Previous study by Piktel, et al. showed that amiodarone did not affect CV, DOR, or action potential duration during ischemia at 36°C. As you know, amiodaron changes various electrophysiological properties in the heart. If you could not examine the effect of amiodaron at BT by yourself, please cite adequate reference to compare the effect of amiodaron in BT and hypothermia.

Response:

Thanks for your great comment. 

In our present study, we investigated the effect of hypothermia and the hypothermia + amiodarone and compared to the baseline temperature. The impact of amiodarone on the baseline temperature is important. 

Studies on amiodarone have revealed that intravenous amiodarone has minimal effect on the refractory periods of the ventricular muscle in baseline condition [Am Heart J. 1984;108(4 Pt 1):890-8; J Am Coll Cardiol. 1986;7(1):148-57] and might decrease conduction velocity in the ventricle. [Circulation. 1995;91(2):451-61; J Am Coll Cardiol. 1985 Jul;6(1):179-85] Previous study by Piktel, et al. showed that amiodarone did not affect CV, dispersion of repolarization, or action potential duration during ischemia at 36°C. [J Am Heart Assoc. 2021 May 18;10(10):e016676, 33938226]

We cited above-mentioned reference add revised the discussion (line 316-322) as follows, “

The impact of amiodarone alone without TH was not investigated in our present study. Studies on amiodarone have revealed that intravenous amiodarone has minimal effect on the refractory periods of the ventricular muscle in baseline condition31,32 and might decrease conduction velocity in the ventricle.33,34 A recent study by Piktel et al.4 showed that amiodarone did not affect CV, dispersion of repolarization, or action potential duration during ischemia at 36°C. Our study echoed the finding of the decrease in the CV after amiodarone infusion even under the TH. 

Reference

4. Piktel JS, Suen Y, Kouk S, Maleski D, Pawlowski G, Laurita KR, Wilson LD. Effect of Amiodarone and Hypothermia on Arrhythmia Substrates During Resuscitation. J Am Heart Assoc. 2021;10:e016676. doi: 10.1161/JAHA.120.016676

31. Ikeda N, Nademanee K, Kannan R, Singh BN. Electrophysiologic effects of amiodarone: experimental and clinical observation relative to serum and tissue drug concentrations. Am Heart J. 1984;108:890-898. doi: 10.1016/0002-8703(84)90451-4

32. Morady F, DiCarlo LA, Jr., Krol RB, Baerman JM, de Buitleir M. Acute and chronic effects of amiodarone on ventricular refractoriness, intraventricular conduction and ventricular tachycardia induction. J Am Coll Cardiol. 1986;7:148-157. doi: 10.1016/s0735-1097(86)80273-x

33. Nanas JN, Mason JW. Pharmacokinetics and regional electrophysiological effects of intracoronary amiodarone administration. Circulation. 1995;91:451-461. doi: 10.1161/01.cir.91.2.451

34. Morady F, DiCarlo LA, Jr., Baerman JM, Krol RB. Rate-dependent effects of intravenous lidocaine, procainamide and amiodarone on intraventricular conduction. J Am Coll Cardiol. 1985;6:179-185. doi: 10.1016/s0735-1097(85)80272-2

“

# 2. You prepared 6 pigs for protocol I, however, there are only data from 3 pigs for the evaluation of vulnerability to ventricular arrhythmias. Why did you discard the data from other 3 pigs?

Response:

Thanks for your great comment. 

Three of the pigs didn’t underwent the evaluation of vulnerability to ventricular arrhythmias.

We revised the limitation as follows, (page 25, line 368-369) as follows,”

The number in animals for vulnerability test for ventricular arrhythmias is small for a solid conclusion.“

# 3. I could not find the detail of protocol II and III. In protocol I, you performed burst right ventricular pacing after amiodaron injection in TH. In protocol II, did you perform burst pacing just after TH, and in protocol III, you just performed burst pacing? Did you measure the baseline characteristics of 4 and 4 pigs in protocol II and III? Were the baseline electrophysiological characteristics of pigs in protocol II and III same with those in protocol I? Please revise the supplemental figure 1B and figure legend for the easy understanding of the readers.

Response:

Thanks for your great comment. 

We performed burst pacing just after TH in the protocol II and in the BT in the protocol III. Baseline characteristics, electrocardiographic parameters, and electrophysiological parameters were not measured in the protocol II and protocol III.

We revised the Supplemental Figure 1 accordingly as follows, “

Legend

(A) In the first part of protocol I, epicardial window was created for the high-density epicardial mapping under sinus rhythm and right ventricular pacing rhythm by using multi-electrode mapping catheter in 6 pigs. Epicardial mapping was repeated after the induction of TH and after the infusion of amiodarone. 

(B) In the second part of protocol I, vulnerability to VA was performed in 3 swine. Another 4 and 4 swine underwent protocol II and protocol III respectively. In the protocol II and III, vulnerability to VA was examined under TH and BT respectively without 3D mapping /electrophysiological study. In the protocol II, induction of TH was performed and vulnerability to VA was conducted in 4 swine after an observation time of 60 minutes. In the protocol III, vulnerability to VA was conducted in 4 swine after an observation time of 120 minutes.

OT = outflow tract; RV = right ventricle; LV = left ventricle; 3D = three-dimensional; VA = ventricular arrhythmias

“

# 4. Why did the inter-segmental wavelet propagation change from BT to TH and return to the same pattern with BT after amiodaron/TH? Please discuss the mechanism in the discussion section.

Response:

Thanks for your great comment. 

We added discussion as follows (line 339-353), “

The inter-segmental wavelet propagation was determinate by CV of each segment. In our present study, we investigated the epicardial CV and electrogram of each segment. We observed a significant reduction in the CV after induction of TH in the anterior mid RV (segment 8), which was greater than that in the other segments, which caused the wavelet propagation change from BT to TH. After the infusion of amiodarone under TH, the CV of each segment was further decreased and alleviated the difference between the anterior mid RV (segment 8) and other segments. This change in CV resulted in the reverse of inter-segmental wavelet propagation inter-segmental wavelet propagation. The mechanism might be related to the change of distribution of connexin 43 or the heterogenous change of CV in the endocardium/transmural myocyte.37 The amiodarone could provide heterogeneous effect on the myocardium and decrease the ventricular repolarization dispersion.38 Our study indicated the amiodarone might provide heterogenous effect on the myocardial CV under TH. Further study investigating the electrophysiological parameter with simultaneous endocardial and epicardial mapping to clarify the is warrant. 

“

Reference 

37. Dietrichs ES, McGlynn K, Allan A, Connolly A, Bishop M, Burton F, Kettlewell S, Myles R, Tveita T, Smith GL. Moderate but not severe hypothermia causes pro-arrhythmic changes in cardiac electrophysiology. Cardiovasc Res. 2020;116:2081-2090. doi: 10.1093/cvr/cvz309

38. Drouin E, Lande G, Charpentier F. Amiodarone reduces transmural heterogeneity of repolarization in the human heart. J Am Coll Cardiol. 1998;32:1063-1067. doi: 10.1016/s0735-1097(98)00330-1

 

# 5. How did you obtain the cardiomyocytes for Immunofluorescent staining and western blotting, biopsy or after sacrifice? How did you treated the heart if you obtain the cardiomyocytes after sacrifice? Did you obtain the cardiomyocytes from endomyocardium or epicardium?

Response:

Thanks for your great comment. 

We obtained the tissue after sacrifice. A fixation procedure was performed immediately in both 20% formalin and liquid nitrogen to prevent sample degradation. We obtained the cardiomyocytes from both endomyocardium and epicardium.

We revised the supplemental file (line 45-49) as follows, 

“

Tissue sampling

We obtained the tissue from the pre-specified 13 segments after sacrifice. A fixation procedure was performed immediately in both 20% formalin and liquid nitrogen to prevent sample degradation. We obtained the cardiomyocytes from both endomyocardium and epicardium.

“

# 6. Please describe how to calculate the total and lateralization of connexin 43 using image J software.

Response:

Thanks for your great comment. 

Assisted by image analysis software, the percentage of total tissue area occupied by connexin 43 immunoreactive signals, and the percentage of connexin 43 signal located outside the end-to-end cell junctions (lateralization). The percentages of lateralization in specific segment were means of the 24 slides (6 slides from segment 5, 8, and 12 of each pig) from the specific segment.

Please see the revised supplemental file, line 66-70.

# 7. The sample numbers in graphs in figure 6A is 24. Did you obtain 6 specimens from each pig? Please add the explanation in the methods.

Response:

Thanks for your great comment. 

Please see the response to the #7. 

# 8. Did you check the connexin 43 levels in pigs assigned to protocol II and III? Conduction velocity was changed after TH or TH/amiodaron therapy. Showing connexin 43 levels in the BT in each segment are important to explain the relationship between conduction velocity change and the expression level of connexin 43.

Response:

Thanks for your great comment. 

We didn’t check the connexin 43 levels in pigs assigned to protocol II and III.

We only examined the vulnerability to ventricular arrhythmias in the protocol II and III.

# 9. In the cardiac conduction system, connexin 45 is highly expressed. Did you evaluate the expression level of connexin 45?

Response:

Thanks for your great comment. 

We realized that Cx45 is one of the most important connexins expressed in the heart and is localized to the sinoatrial node, atrioventricular node, and His-Purkinje system, but has also been shown to be expressed in atrial and ventricular myocytes. [Curr Opin Cardiol, 30 (2015), pp. 33-39] In our present study, we didn’t evaluate the expression levels of connexin 45. The further study investigating both the expression of Cx45 and Cx43 may enrich out study and improve the quality. We will work on this. Thanks for your valuable comments!

---

## [Decision Letter · Decision Letter 1]

28 Feb 2023

Epicardial electrical heterogeneity after amiodarone treatment increases vulnerability to ventricular arrhythmias under therapeutic hypothermia

PONE-D-22-27467R1

Dear Dr. HU,

We’re pleased to inform you that your manuscript has been judged scientifically suitable for publication and will be formally accepted for publication once it meets all outstanding technical requirements.

Kind regards,

Tomohiko Ai, M.D., Ph.D.

Academic Editor

PLOS ONE

Additional Editor Comments (optional):

Reviewers' comments:

Reviewer's Responses to Questions

**Comments to the Author**

1. If the authors have adequately addressed your comments raised in a previous round of review and you feel that this manuscript is now acceptable for publication, you may indicate that here to bypass the “Comments to the Author” section, enter your conflict of interest statement in the “Confidential to Editor” section, and submit your "Accept" recommendation.

Reviewer #1: All comments have been addressed

2. Is the manuscript technically sound, and do the data support the conclusions?

Reviewer #1: Yes

3. Has the statistical analysis been performed appropriately and rigorously? 

Reviewer #1: Yes

4. Have the authors made all data underlying the findings in their manuscript fully available?

Reviewer #1: Yes

5. Is the manuscript presented in an intelligible fashion and written in standard English?

Reviewer #1: Yes

6. Review Comments to the Author

Reviewer #1: Authors replied to my previous comments adequately. I have no additional comments to this manuscript.

7. PLOS authors have the option to publish the peer review history of their article (what does this mean?). If published, this will include your full peer review and any attached files.

Reviewer #1: No

---

## [Editor Report · Acceptance letter]

11 Apr 2023

PONE-D-22-27467R1 

Epicardial electrical heterogeneity after amiodarone treatment increases vulnerability to ventricular arrhythmias under therapeutic hypothermia 

Dear Dr. Hu:

I'm pleased to inform you that your manuscript has been deemed suitable for publication in PLOS ONE. Congratulations! Your manuscript is now with our production department. 

Kind regards, 

on behalf of

Dr. Tomohiko Ai 

Academic Editor

PLOS ONE